# Removal of Synthetic Dye from Aqueous Solution Using MnFe₂O₄-GO Catalyzed Heterogeneous Electro-Fenton Process

**Gayathri Anil** [1,2], **Jaimy Scaria** [2] **and Puthiya Veetil Nidheesh** [2,*] 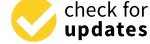

1   Sacred Heart College, Thevara, Kochi 682013, India
2   CSIR-National Environmental Engineering Research Institute, Nagpur 440020, India
*   Correspondence: pv_nidheesh@neeri.res.in or nidheeshpv129@gmail.com

**Abstract:** In the present study, heterogeneous electro-Fenton (HEF) process using MnFe₂O₄-GO catalyst is employed for the successful removal of dye from aqueous solution. Pt coated over titanium and graphite felt were used as the electrodes. The study focuses on the efficiency of the electrodes and catalyst used for the successful removal of Rhodamine B (RhB) from aqueous solution and the application of the same in real textile wastewater. The effect of various operational parameters like pH, applied voltage, catalyst concentration, initial pollutant concentration and effect of ions were investigated. The optimized condition of the electrolytic system was found as pH 3, applied voltage of 3 V, and catalyst concentration of 20 mg L$^{-1}$ for the removal of 10 ppm RhB. At the optimized condition, 97.51% ± 0.0002 RhB removal was obtained after an electrolysis time of 60 min. The role of individual systems of Fe, Mn, GO and MnFe₂O₄ without support were compared with that of catalyst composite. On examining the practical viability in real textile effluent, a significant colour reduction was observed (reduced by 61.24% ± 0.0261 in 60 min). Along with this, the biodegradability enhancement (BOD/COD ratio from 0.07 to 0.21) after treatment was also observed.

**Keywords:** heterogeneous electro-Fenton; composite catalyst; Rhodamine B dye; textile wastewater; biodegradability; advanced oxidation processes

## 1. Introduction

Textile effluent, defined by a solid colour, soluble dyes, and organics, makes up half of all industrial wastewater in the world [1]. India stands second in the world in the production of textiles and fabrics [2]. Additives used in the textile industry for various softening and polishing purposes include colouring pigments, such as dyes and surfactants [3]. Synthetic dyes in the textile effluent maintain their colour and structural integrity under extreme weather situations and exhibit a strong resistance to microbial degradation [4,5]. They take a long time to degrade when exposed to the environment. These toxic organic pollutants generated from textile industries can affect living beings and can cause serious threats to humans and the environment. Synthetic dyes cause a wide range of health effects on living beings. They impart colour to the aquatic environment, even in a small concentration [6]. Synthetic dyes possess carcinogenic, mutagenic and genotoxic properties, prolonged exposure to which can cause human health effects, including skin/eye irritation, neurotoxicity, and endocrine disruption [7,8]. Based on the types of dyes and accompanying chemicals employed, the behavior of textile discharge changes [9]. Textile wastewater usually has a low BOD to COD ratio (less than 0.1) making it unfit for biodegradation [10]. The effluent has to be treated effectively before discarding into the environment [11].

Electrochemical advanced oxidation processes (EAOPs) are one of the most environmentally benign processes for wastewater treatment, because the reagent utilized is the electron, and the oxidizing agents, such as hydroxyl radical, are in-situ generated [12]. The process is considered to be very effective since it is ecologically friendly and can create good amount of hydroxyl radicals by controlling the applied current [13]. It is considered as an

effective method for the treatment of dyes, as the dye compounds are readily converted into carbon dioxide, water and other inorganic ions [5].

The electro-Fenton (EF) process is the most accepted EAOP, due to the benefits such as a broad application range, excellent anti-interference ability, ease of operation, and rapid pollutant removal and mineralization efficiency [14–16]. For the past few years, EF has been the most employed EAOP method and is considered most effective due to its ability to eliminate refractory compounds [17,18]. The efficacy of the EF procedure as a tool for removing dyes from water medium has been established [19].

The EF system is supplied continuously with air or oxygen and $H_2O_2$ is in-situ formed by the two-electron reduction of oxygen (Equation (1)) at the cathode. The $^\bullet OH$ are generated in the system by the Fenton reaction (Equation (2)), by the electrogenerated $H_2O_2$ at the cathode and $Fe^{2+}$ catalyst added to the system [20]. The $Fe^{2+}$ that is used by the system is then regenerated by the cathodic reduction of $Fe^{3+}$ (Equation (3)) [21,22]. The formation of OH takes place, preferably in acidic media [23,24].

$$O_2 + 2H^+ + 2e^- \rightarrow H_2O_2 \tag{1}$$

$$Fe^{2+} + H_2O_2 \rightarrow Fe^{3+} + {}^\bullet OH + OH^- \tag{2}$$

$$Fe^{3+} + e^- \rightarrow Fe^{2+} \tag{3}$$

The EF process can be classified into two types, based on the catalyst used. The process can be said to be a homogenous electro-Fenton process, if the externally added catalyst is soluble in the system. If the catalyst is not soluble or is barely soluble in the system, the process is known as heterogeneous electro-Fenton (HEF). In a homogeneous system, the catalytic process takes place in the bulk liquid phase, whereas the catalytic process occurs on the surface of the catalyst in a heterogeneous system [25,26]. The mechanism of both the process is more or less similar. In HEF, the solid iron or its minerals helps in the generation of $^\bullet OH$ [27].

The main advantage of HEF process over conventional EF process is that, the heterogeneous catalyst can be recovered and reused after the Fenton process. This lowers the cost of operation and reduces catalyst wastage. The HEF process could work in an extended range of pH. This way adjusting the pH of the solution in the conventional Fenton process can be avoided. All the reactions associated with the process occur at the catalyst surface. The ferric or ferrous ion which is present in the catalyst do not form hydroxyl complexes as they are mostly in the stable form. The production of iron sludge is also reduced by the HEF process [28].

HEF employing iron-based clays, zero-valent iron, iron oxide minerals and iron-containing materials, has generated a lot of interest. The material must have robust catalytic activity that is unaffected by experimental conditions, particularly pH, as well as high stability to ensure reusability across several runs [17]. The most extensively used catalysts are hematite, goethite, and magnetite because of their specific properties [29]. Iron oxides are considered as environmentally acceptable Fenton catalyst, as they are abundant in nature and are easy to synthesize, making them low cost [23]. Recently, various transition metals such as cobalt, copper, nickel, vanadium and manganese have been utilized as solid catalysts with iron/iron oxides [17]. Adding transition metals such as Cu, Mn and Ni in $Fe_3O_4$ is reported to be effective for the increased catalytic activity [30]. Since graphene has been discovered to considerably enhance catalyst activity, researchers have hybridized metal oxides with graphene for its immense supporting power [31]. In the present study, an approach has been made to combine the iron and transition metals with a support of graphene oxide (GO) to develop a heterogeneous catalyst and to examine its efficiency for the removal of RhB dye. For the remediation of various pollutants, such as pharmaceuticals, etc., the combination of these has shown to be successful [32,33]. When manganese was present in a magnetite structure, the rate of $H_2O_2$ decomposition for the breakdown of organic molecules increased noticeably [34,35].

A number of studies have been conducted on the removal of RhB from aqueous solution using HEF. Jinisha et al. [36] used iron-doped SBA-15 mesoporous silica as a heterogeneous catalyst for the effective removal of RhB from aqueous solution. The authors used graphite felt as both anode and cathode. Tian et al. [37] used sponge iron as the catalyst along with gas diffusion electrodes (GDE) for RhB removal. Zheng et al. [38] studied the removal of RhB from aqueous solution using $NiFe_2O_4/Fe_2O_3$ (a charcoal shaped catalyst) as the heterogeneous catalyst and graphite as the electrodes. The importance of Mn based ferrites for HEF is found nowhere in the literature.

This study aims to develop $MnFe_2O_4$-GO as the heterogeneous EF catalyst for the effective removal of RhB dye from aqueous solution. Pt coated over titanium was used as the anode and graphite felt was used as the cathode for the in-situ generation of $H_2O_2$. The effects of various operational parameters including pH, applied voltage, catalyst concentration, pollutant concentration, and the effect of ions were studied and the process was optimized. The optimized conditions were applied to the real textile wastewater and the rate of removal was investigated.

## 2. Materials and Methods

### 2.1. Chemicals and Reagents

Analytical grade chemicals including RhB ($C_{28}H_{31}ClN_2O_3$) supplied from Loba Chemie (Mumbai, India) were used for the experiments. Ferric chloride anhydrous ($FeCl_3$, ≥98%) purchased from Merck (Mumbai, India), manganese chloride ($MnCl_2 \cdot 4H_2O$, ≥ 95%) purchased from Sisco Research Laboratory (Mumbai, India), sodium acetate ($CH_3 \cdot COONa \cdot 3H_2O$, 99.0–101.0%) purchased from Qualigens were used for the synthesis of $MnFe_2O_4$ catalyst. Graphite powder (99.5%) for the synthesis of graphene oxide (GO), was purchased from S D Fine Chemicals (Chennai, India).

Other chemicals, including potassium dichromate ($K_2Cr_2O_7$, ≥99.0%), potassium dihydrogen phosphate ($KH_2PO_4$, ≥99.5%), di-potassium hydrogen phosphate ($K_2HPO_4$, ≥99.0%), ammonium chloride ($NH_4Cl$, ≥99.0%), sodium thio sulphate ($Na_2S_2O_3$, ≥99.5%), t-butanol($(CH_3)_3COH$, ≥99%), sodium hydroxide pellets (NaOH, ≥ 97%), sodium azide ($NaN_3$, ≥99%), Benzoquinone (≥98%), Phenanthroline ($C_{12}H_8N_2 \cdot H_2O$, ≥99.5%), copper sulphate ($HgSO_4$, ≥99.5%), silver nitrate ($AgNO_3$, ≥99.5%), silver sulphate ($AgSO_4$, ≥98%), magnesium sulphate ($MgSO_4$, ≥98%), sodium sulphate ($Na_2SO_4$, ≥99.0%), potassium permanganate ($KMnO_4$, ≥99.0%), and manganese sulphate ($MnSO_4 \cdot H_2O$, ≥98%), were procured from Merck. Sodium chloride (NaCl, ≥99.5%), sodium bicarbonate ($Na_2CO_3$, ≥99.9%), di sodium hydrogen phosphate ($Na_2HPO_4 \cdot 7H_2O$, ≥99.9%), magnesium sulphate ($MgSO_4 \cdot 7H_2O$, ≥99.9%), calcium chloride ($CaCl_2$, ≥98%), ferric chloride ($FeCl_3 \cdot 6H_2O$, ≥98%), ferrous ammonium sulphate ($(NH_4)_2(SO_4)_2 \cdot 6H_2O$, ≥99%), were procured from Fisher Scientific. Ammonium acetate ($CH_3COONH_4$, ≥98%), hydroxyl amine ($NH_2OH$, ≥96%), and potassium iodide (KI, ≥99%) were purchased from Qualigens.

Solvents including ethylene glycol ($HOCH_2$- $CH_2OH$, ≥99.0%), glycerol and ethyl alcohol ($C_2H_5OH$, ≥95%) from Merck, and poly-ethylene glycol (99.9%) from Qualigens, were used for different analytical experiments. Sulphuric acid ($H_2SO_4$, 95.0–98%) and ferroin indicator from Merck, and phenolphthalein supplied from Rankem, were used for the experiments.

### 2.2. Electrodes

Pt coated over titanium plates procured from Titanium Tantalum Products Limited, Chennai, India, was used as an anode and graphite felt was used as a cathode.

### 2.3. Preparation of Heterogeneous Catalyst

#### 2.3.1. Synthesis of Graphene Oxide (GO)

For the synthesis of graphene oxide (GO) from natural graphite powder, a modified version of Hummers' method combined with laboratory optimizations, as mentioned in our earlier work [39], was used. In a water bath, 1 g of the graphite powder was mixed

with 68 mL of conc. $H_2SO_4$ and mixed thoroughly for 1 h. While keeping the ice bath conditions (to keep the temperature below 20 degrees), 4 g $KMnO_4$ was added gradually (1 g at a time). The resultant mixture was agitated at room temperature for 6 h, until the solution changed colour from green to dark brown–black. The oxidation was then stopped by drop-wise addition of 140 mL of double-distilled water (DDW), which followed by 5 mL $H_2O_2$ addition. Finally, the metal ions were removed by washing the mixture three times with 10% HCl.

Washing with DDW was repeated (by centrifugation at 7500 rpm) until the pH of the supernatant liquid approached neutral condition. The addition of silver nitrate confirmed the removal of chloride from the supernatant. After drying at 40 °C in a hot air oven, the resultant solid was pulverized [39].

### 2.3.2. Synthesis of Manganese Ferrite Supported on GO

To synthesize $MnFe_2O_4$–GO composite, the facile solvothermal reduction was used. To a 250 mL volumetric flask containing 40 mL ethylene glycol (EG), 0.0387 g of GO was added. The mixture was kept in a bath sonicator for 1 h for the effective dispersion of GO. 0.811 g ferric chloride and 0.3010 g manganese chloride was added. The mixture was kept for magnetic stirring between 650–700 rpm until the $FeCl_3$ and $MnCl_2$ added were completely dissolved. A total of 3.6 g sodium acetate ($CH_3.COONa.3H_2O$) followed by 1 mL polyethylene glycol (PEG) were added to the above mixture and agitated at 550 rpm for 1 h. The agitated mixture was then transferred to a Teflon-lined stainless steel autoclave and kept for 6 h at 200 °C. After cooling, the precipitate was then separated using a magnet. The dark brown precipitate was washed thrice with ethanol and DDW. The washed precipitate was then vacuum dried for 6 h. After cooling, it was powdered and used for experiments [39]. EG used in the preparation acted as both solvent and reducing agent [39].

To find out the role of Fe, Mn, GO and spinel structure in the catalyst, the above method was followed. Catalysts were prepared by adding Fe only ($MnCl_2$ was not added), Mn only ($FeCl_3$ was not added), GO only (both $FeCl_3$ and $MnCl_2$ was not added) and $MnFe_2O_4$ without support (GO was not added).

### 2.4. Catalyst Characterization

The structural characterization of the laboratory-synthesized catalyst was done using X-ray powder diffraction (XRD, Bruker D8 Advance). The functional groups present on the surface of the catalyst were detected by Fourier transform infrared (FTIR) spectroscopy (FTIR, Bruker, Germany). The size of the prepared catalyst was determined by transmission electron microscope (TEM, Tecnai G2 20 S-TWIN).

### 2.5. Experimental Procedure

Electrolytic experiments were carried out in a reactor of 1 L capacity. A 1000 mL solution containing of 10 ppm RhB was prepared at the beginning of each experiment. Initial pH was maintained at 3 for all the experiments, except those in which the effect of different pH was examined. pH was adjusted using 0.1 N $H_2SO_4$ and 0.1 N NaOH. pH was monitored using a Laqua pH meter (Horiba scientific). Two anodes and two cathodes of size 5 × 8 cm with a 1 cm inner electrode spacing were used for the experiments. Using a commercially available fish aerator, a continuous supply of air was fed into the system until the end of the experiments. After an aeration time of 5 min, catalyst was added. The electrodes were connected to a DC power supply (make: Crown). The experimental setup is shown in Figure 1. Samples were collected at 15 min intervals and stored in the refrigerator in brown vials to avoid further degradation or reaction. The residual RhB present in each sample was measured using a UV/Vis spectrophotometer (UV-1900I, Shimadzu, Japan) at 555 nm (which is the peak wavelength of RhB). All the experiments were conducted at room temperature. The effect of different operational parameters such as pH, voltage, catalyst concentration, and pollutant concentration were also studied.

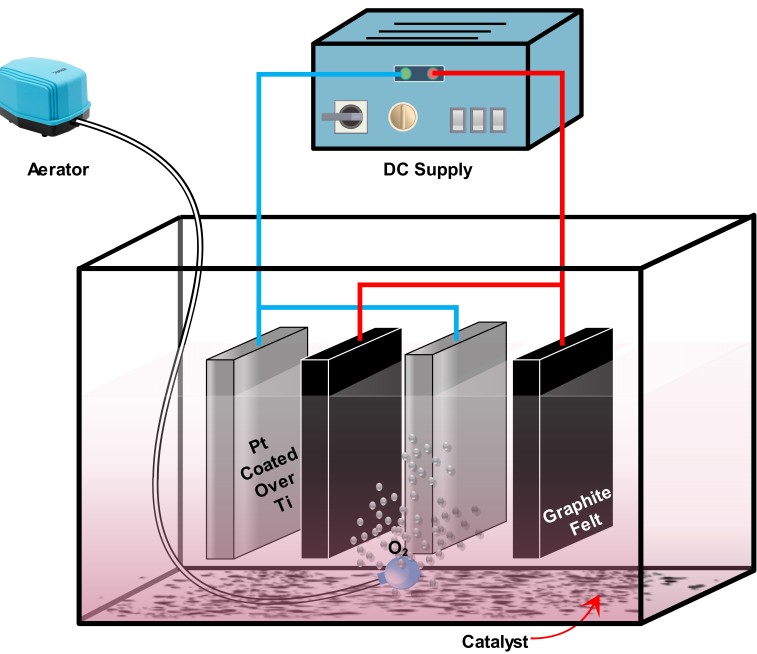

**Figure 1.** Schematic diagram of the experimental setup.

## 2.6. Real Textile Wastewater: Collection and Analysis

Real textile wastewater was collected from an industry situated in Mumbai, Maharashtra, India. A 10 L sample was collected, sealed tightly and stored in the refrigerator. Parameters including total dissolved solids (TDS), total solids (TS), chloride, sulphate, biological oxygen demand (BOD), chemical oxygen demand (COD) were analyzed as per the standard methods [40]. pH and electrical conductivity of the wastewater at room temperature were recorded using a Laqua pH meter (Horiba scientific). The absorbance of the wastewater was recorded using a UV/Vis spectrophotometer (UV 1900 I, Shimadzu, Japan). Using a total organic carbon (TOC) analyser (Shimadzu, Japan, Model: TOC Veph), the TOC content of the real textile wastewater was determined.

## 2.7. Energy Consumption

The specific energy consumption with different applied voltages was found using the equation mentioned in [41].

$$Q = \frac{VIt}{m} \tag{4}$$

where Q = specific energy consumption $(KWh^{-1})$, V = applied voltage (V), I = current (A), t = electrolysis time (min), m = change in concentration.

## 2.8. Leaching of Catalyst

The quantity of Fe leached was quantified using a 1.10-phenanthroline method by spectrophotometric quantification. The quantity of Mn leached from the catalyst was quantified using ICP-OES (iCAP-7400, Thermo Fisher Scientific, Waltham, MA, USA).

## 3. Results and Discussions

### 3.1. Characterization of the Catalyst

XRD, TEM and FTIR were used to characterize the $MnFe_2O_4$–GO catalyst. XRD patterns provide the crystal structure and phase information of the catalyst [42]. Figure 2a illustrates the XRD pattern of $MnFe_2O_4$–GO. The obtained diffraction peaks are similar to that of JCPDS 38-0430. The peaks observed at 2 θ values of 16.92, 29.59, 35.75, 36.32, 42.92, 52.59, 56.01, 63.96, 76.64, which can be indexed as (111), (202), (311), (222), (004), (422), (333), (440), and (533) planes, respectively. Similar patterns of $MnFe_2O_4$ were ob-

served earlier [42,43]. The peak observed at the 2 θ value of 7.69 corresponds to the (001) characteristic plane of GO [44]. The presence of a GO peak indicates the ethylene glycol is insufficient to convert GO to rGO when more than one metal oxide is present during the solvothermal process, whereas this conversion is viable when only one metal oxide is present, as reported in our previous study [39]. This may be because the amount of solvent and time taken for the synthesis may not be enough for the reduction of GO [45].

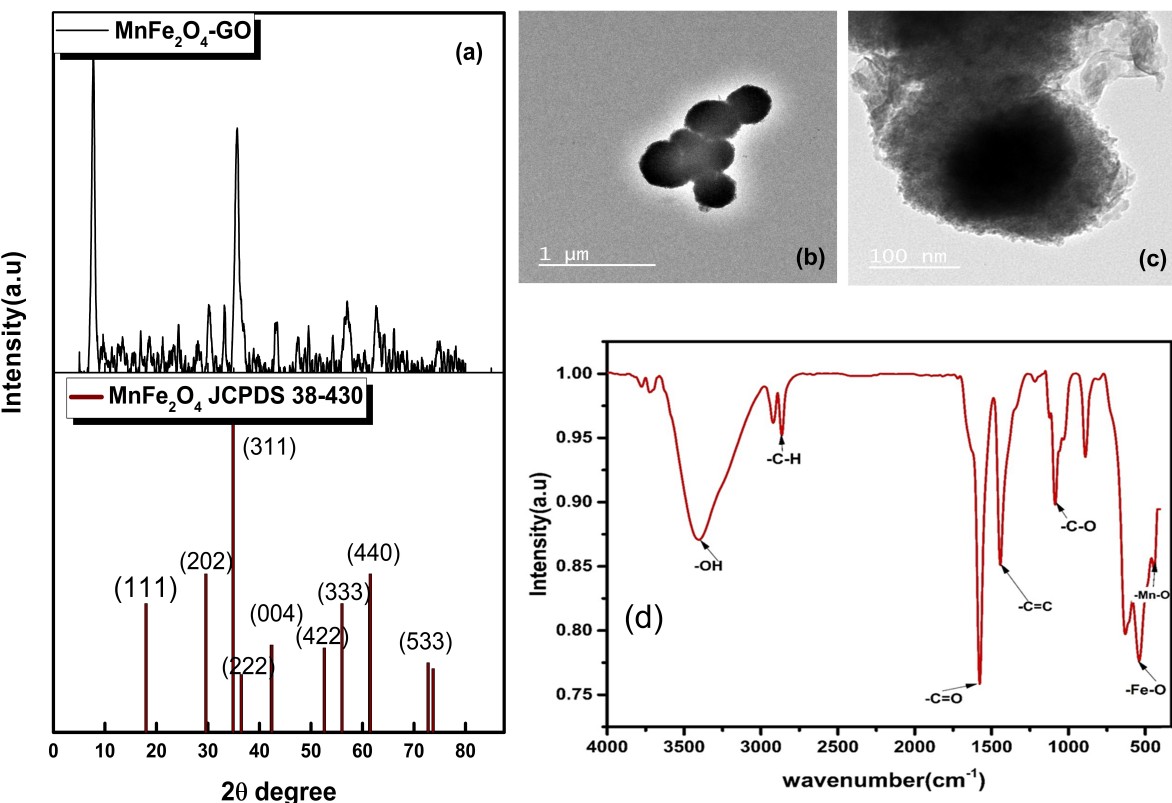

**Figure 2.** XRD pattern of $MnFe_2O_4$–GO (**a**) TEM images of $MnFe_2O_4$–GO (**b**), (**c**), FTIR spectrum of $MnFe_2O_4$–GO (**d**).

The shape, size and distribution of particles can be found using a TEM [46]. Figure 2b,c shows the TEM images of the catalyst composite synthesized. From Figure 2b,c, the irregular rough surface $MnFe_2O_4$ can be observed, indicating the small $MnFe_2O_4$ particles are clustered together to form a spherical aggregate, with a diameter of $132 \pm 92$ nm. The presence of GO sheets is evident in Figure 2c. On further examination of Figure 2b,c, the different stages of spherical structure development indicate that aggregation and particle growth are not completed and are at different phases.

The FTIR spectrum of $MnFe_2O_4$ -GO is shown in Figure 2d. As illustrated in the figure, bands of functional groups such as –OH, –C=O, C– H, C–O–C/C–O–H, –C=C, and –C–O were obtained at around 3403.9 $cm^{-1}$, 2917.8 $cm^{-1}$, 1581.6 $cm^{-1}$, 1125.5/1437.3 $cm^{-1}$, and 1081.3 $cm^{-1}$, respectively. The bands observed at 580 $cm^{-1}$ and 443 $cm^{-1}$ can be attributed to the bonds corresponding to Fe-O and Mn-O, respectively [47,48]. The presence of oxygen functionalities implies the presence of GO [45].

### 3.2. Rhodamine B Degradation in $MnFe_2O_4$–GO-Based Electro-Fenton System

3.2.1. Comparison between Different Processes

During HEF treatment of dye, a combination of various processes such as adsorption, anodic oxidation (AO), and AO+$H_2O_2$ may be contributed. In order to determine the contribution of these processes, degradation of RhB in different individual processes was carried out. Adsorption of dye on catalyst was analyzed in the absence of current supply

and aeration. The effect of AO was investigated by applying electricity to the electrochemical system without providing catalyst and aeration. AO + $H_2O_2$ was carried out in the electrochemical system with electrodes, and external aeration without the addition of the catalyst. The other experimental conditions were pH 3, applied voltage of 3 V and initial pollutant concentration of 10 ppm.

Figure 3a depicts the degradation rate achieved with different processes. The different degradation percentage obtained by the individual process is as follows: 12.36% $\pm$ 0.0028 by adsorption, 64.91% $\pm$ 0.0439 by AO, 90.42% $\pm$ 0.0395 by AO + $H_2O_2$ and 97.52% $\pm$ 0.0002 by HEF process. On examining the kinetics, all the above individual process follows first-order kinetics. The first-order rate constant of each process are as follows: adsorption (0.0024 $min^{-1}$), anodic oxidation (0.019 $min^{-1}$), anodic oxidation combined with aeration (0.04 $min^{-1}$) and HEF (0.079 $min^{-1}$). Even though AO + $H_2O_2$ and HEF showed comparable dye removal within 60 min, the reaction rate of HEF is found to be double than that of AO combined with aeration. Thus, the addition of catalyst can significantly enhance the pollutant removal in EF.

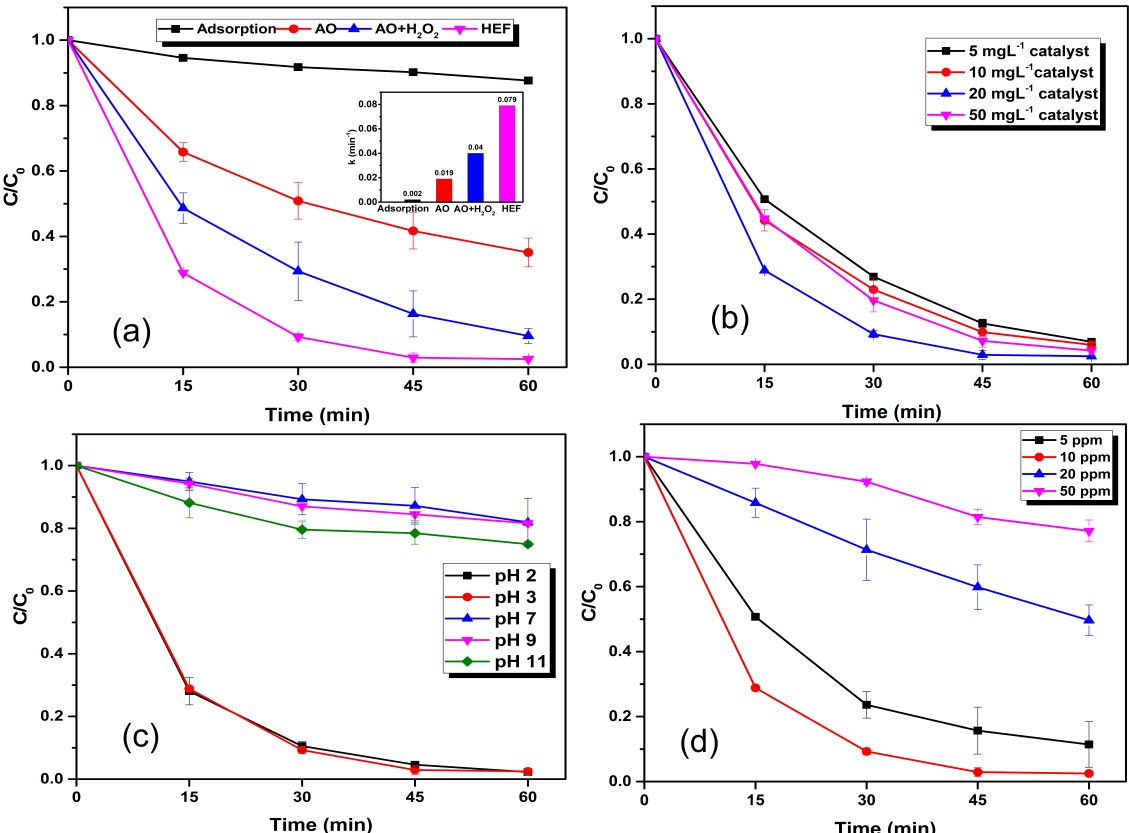

**Figure 3.** (**a**) Degradation of RhB with different process with rate of reaction in inset (**b**) Effect of catalyst dosage (**c**) Effect of pH (**d**) Effect of Initial dye concentration.

Similar results were obtained in HEF treatment of tannery wastewater [49]. The author used $Fe_3O_4$/$Mn_3O_4$/ZnO–rGO hybrid quaternary nano-catalyst. Graphite was used as the cathode and Ti/$IrO_2$/$RuO_2$ as the anode. The authors compared different processes of HEF, AO + $H_2O_2$, AO and adsorption. Higher degradation was obtained by HEF with 97.08% removal. A study by Jinisha et al. [36] compared electro-sorption, adsorption and HEF process. Higher degradation was shown by HEF (97.7%) process. A study conducted by Bedolla-Guzman and co-workers [50] compared the efficiency of AO + $H_2O_2$ and EF process in reactive yellow 160 dye with boron-doped diamond (BDD) as the anode and carbon polytetrafluorethylene (PTFE) as the cathode with $Fe^{2+}$ as catalyst. The authors achieved higher efficiency with the EF process (91%). The study on the degradation of

sunset yellow FCF azo dye [51] revealed that on comparison of AO−H$_2$O$_2$ and EF, the authors achieved higher degradation with EF. Only 88% dye was removed by AO−H$_2$O$_2$ in 360 min, whereas in EF around 50% dye was removed within 5 min, which fully decolorized in 45 min.

### 3.2.2. Effect of Catalyst Dosage

The catalyst dosage has a crucial role in the HEF process [52]. In order to optimize the catalyst dosage, HEF treatment was carried out by varying the catalyst dosage (5 mg L$^{-1}$, 10 mg L$^{-1}$, 20 mg L$^{-1}$ and 50 mg L$^{-1}$). As illustrated in Figure 3b, on increasing the catalyst dosage from 5 mg L$^{-1}$ to 10 mg L$^{-1}$, RhB removal slightly improved from 93.09% ± 0.0081 to 94.02% ± 0.0123. As, Fe$^{2+}$ generation is less at lower catalyst concentrations, the production of hydroxyl radical is also less, which explains the lesser efficiency in degradation at 5 mg L$^{-1}$ catalyst concentration [23,41,53]. When increasing catalyst dosage to 20 mg L$^{-1}$, a sharp increase in pollutant removal was evident, as RhB removal reached 97.51% ± 0.0002 within 60 min. Whereas, further increase in catalyst dosage lowered the RhB removal (reduced to 95.81% ± 0.0066).When excessive catalyst concentration is present in the system, the generated hydroxyl radicals could be scavenged and could lead to hydroperoxyl radical formation which has less oxidation potential than •OH (Equations (5)–(7)) [23,53,54]. Thus, the optimized catalyst concentration is taken as 20 mg L$^{-1}$ for all further experiments.

$$Fe^{2+} + HO^{\bullet} \rightarrow Fe^{3+} + OH^{-} \tag{5}$$

$$Fe^{2+} + H_2O_2 \rightarrow Fe-OOH^{2+} + H^{+} \tag{6}$$

$$Fe-OOH^{2+} \rightarrow HO_2^{\bullet} + Fe^{2+} \tag{7}$$

### 3.2.3. Effect of pH

The solution pH is a decisive parameter of the Fenton process [55]. An acidic pH, preferably pH 3, results in more radical generation, thus more activity. To determine the optimum pH for the HEF process, experiments were carried out under different pH of 2, 3, 7, 9, and 11. As Figure 3c indicates, higher RhB degradation is evident at acidic conditions (2 and 3). At pH 2, RhB removal of 97.72% ± 0.0021 and at pH 3, removal of 97.52% ± 0.0002, were observed. Whereas at neutral pH, only 18.08% ± 0.0765 of RhB was removed. At pH 9, 18.37% ± 0.0082 and at pH 11, 25.06% ± 0.0077 removal, were obtained.

Many studies reported the higher degradation of compounds at pH 3. Zheng et al. [38] observed a high degradation in RhB at pH of 3 on using NiFe$_2$O$_4$/Fe$_2$O$_3$ as the heterogeneous catalyst with graphite felt as the electrodes. Nidheesh et al. [54] studied the degradation of RhB using magnetite as the catalyst and graphite as the electrode. The authors also obtained higher efficiency at the pH of 3. Fayazi and Ghanei-Motlagh, [56] obtained higher degradation efficiency of methylene blue at pH 3 when sepiolite/pyrite composite was used as the catalyst, and graphite and platinum sheet was used as the electrodes. On the degradation of Ponceau SS dye using heterogeneous electro-Fenton process, dos Santos et al. [57] observed higher degradation at pH 3, when vermiculite was used as the catalyst.

For pH above 5, precipitation of ferric oxyhydroxide (FeOOH$^{2+}$) and ferric hydroxide (Fe(OH)$_3$) by ferric ions takes place terminating the Fenton reaction [58,59]. Formation of ferric hydroxides of iron species occurs at higher at higher pH values leading to the lowering of Fe$^{2+}$/Fe$^{3+}$ ratio [23,60,61]. For the formation of H$_2$O$_2$, H$^+$ is needed (Equation (1)). H$^+$ is available only in acidic conditions. So, pH 3 is taken as optimum pH for further experiments.

### 3.2.4. Effect of Voltage

For achieving better process efficiency and for the formation of Fenton reagent, applied voltage or current density is a crucial parameter [12]. The effect of different applied voltages ranging from 1 V to 5 V were scrutinized to find out the optimum voltage. As the

applied voltage was increased, the degradation also increased and after a certain range the degradation rate decreased. When the applied voltage was 1 V, the removal percentage was 22.15% ± 0.0414; on increasing the applied voltage to 2 V, degradation enhanced to 37.32% ± 0.0856. An upsurge degradation was achieved when the applied voltage was 3 V i.e., 97.52% ± 0.0002. On applying a voltage of 4 V, a similar removal percentage of 3 V was obtained (97.44% ± 0.0009) and was further decreased on applying 5 V (68.21% ± 0.0672). The RhB removal attained with different applied voltages is illustrated in Figure 4a.

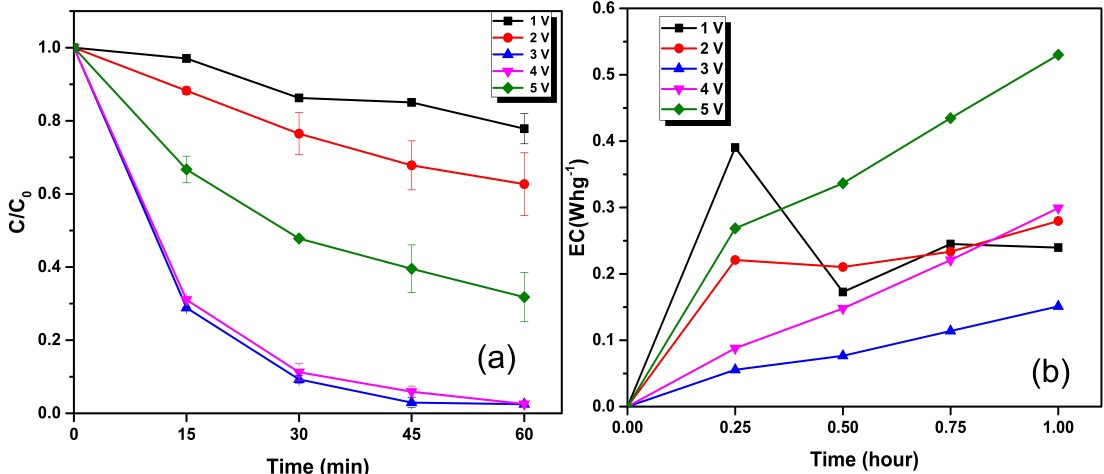

**Figure 4.** (**a**) Effect of voltage on RhB degradation. (**b**) Energy consumption with varying voltage.

The enhancement in degradation with increasing voltage is due to the increase in the production of $^{\bullet}$OH [8,54]. Reduction in oxidation at higher voltages after a certain level is due to the hydrogen gas evolution, decomposition of $H_2O_2$ (Equation (8)) and formation of $H_2O$.

$$2H_2O_2 \rightarrow 4H^+ + O_2 + 4e^- \tag{8}$$

The energy consumption for the RhB removal via HEF is calculated for varying applied voltages (Figure 4b). On comparing different voltages, 3 V showed significant reduction in RhB concentration by utilizing the lower energy. In both higher and lower voltage conditions, energy efficiency was less. Higher energy consumption will increase the cost of the process and thus a voltage of 3 V was selected for further experiments [54].

### 3.2.5. Effect of Initial Dye Concentration and Electrolysis Time

The effect of RhB concentration in HEF activity was evaluated by changing the initial pollutant concentration of 5 ppm, 10 ppm, 20 ppm and 50 ppm. The different removal rates with different concentrations of initial dye are explained in Figure 3d. For 5 ppm dye, 88.57% ± 0.0699 removal was observed which enhanced to 97.51% ± 0.0002 when the dye concentration was increased to 10 ppm. Further increase in concentration affected the efficiency drastically. When the initial dye concentration was 20 ppm, 50.36% ± 0.0472 removal was obtained, which lowered to 22.82% ± 0.0335 when the initial dye concentration was raised to 50 ppm.

Rate of removal of dye decreases with an increase in dye concentration [62]. The decrease in removal of dye with an increase in concentration can be ascribed to the lesser number of available $^{\bullet}$OH for the oxidation [25]. The lower pollutant removal in case of initial dye concentration below 10 ppm is due to the insufficient radical generation. This results in lower dye removal as collision between the particles is limited [54].

RhB degradation readily increases with electrolysis time. From Figure 3d, it is clear that a good amount of RhB particles have been degraded within the first 15 min of the electrolysis. As the time increased, around 97.51% ± 0.0002 was degraded from the system at the end of 60 min. Nidheesh et al. [54,63] reported the increasing RhB degradation with an increase in time. The decolorization rate was higher at the initial time intervals of

electrolysis because of the higher formation of $^{\bullet}$OH. The degradation rate decreased as the time increased, because of the decrease in collision of RhB molecules with $^{\bullet}$OH [54,63].

### 3.2.6. Effect of Ions

The effect of different anions such as chlorides, sulphates and carbonates on RhB solution at optimized conditions was studied. Figure 5 illustrates the effect of various ions on the degradation of RhB. 50 mg L$^{-1}$ and 100 mg L$^{-1}$ of each NaCl, Na$_2$SO$_4$ and Na$_2$CO$_3$ were added to the RhB solution.

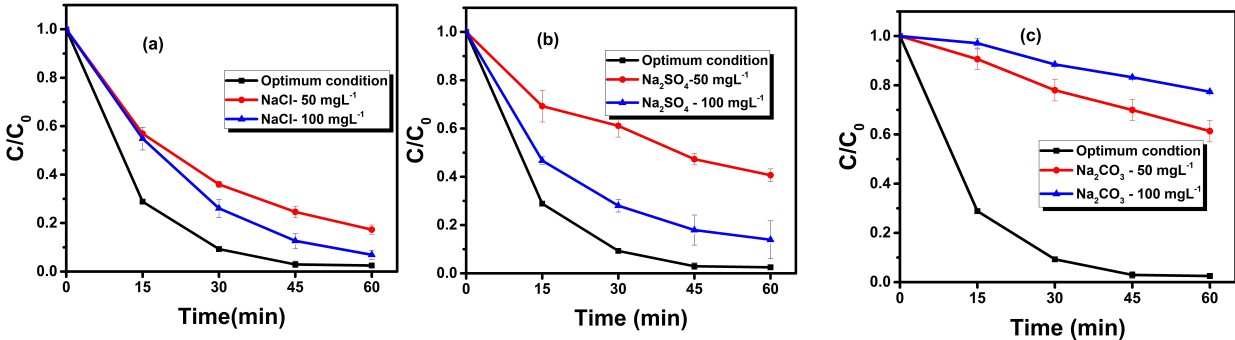

**Figure 5.** (**a**) Effect of NaCl on the degradation of RhB. (**b**) Effect of Na$_2$SO$_4$ on the degradation of RhB. (**c**) Effect of Na$_2$CO$_3$ on the degradation of RhB Experimental conditions: pH-3, Voltage–3, Catalyst dosage- 20 mg L$^{-1}$, Initial dye concentration-10 ppm.

When NaCl was added into the RhB solution, the removal percentage was 68.21% $\pm$ 0.0189 for 50 mg L$^{-1}$ of NaCl and 93.1% $\pm$ 0.0184 for 100 mg L$^{-1}$ of NaCl. When 50 mg L$^{-1}$ of Na$_2$SO$_4$ was added to the RhB solution the degradation percentage obtained was 59.35% $\pm$ 0.0265, and on addition of 100 mg L$^{-1}$ the degradation obtained was 86.1% $\pm$ 0.0782. When 50 mg L$^{-1}$ of Na$_2$CO$_3$ was added, the degradation percentage obtained was 38.63% $\pm$ 0.0437, and 22.63% $\pm$ 0.0039 when 100 mg L$^{-1}$ of Na$_2$CO$_3$ was added. Among the different anions added, the presence of carbonates has shown a strong influence in HEF activity. Whereas other ions such as chloride and sulphate inhibit HEF activity at lower concentration, and on higher dosage, can result in supporting the HEF activity via additional radical generation.

The presence of Cl$^-$ in wastewaters have a suppressing effect on AOPs as they are said to have scavenging effect on $^{\bullet}$OH [64]. However, in this case, addition of chlorides as well as sulphates does not diminish the degradation but on the contrary increased the degradation with the increase of concentration. The upgrade in degradation with the increase of sulphate concentration was reported by Zhou et al. [65]. The authors explained the increase in degradation as the enhancement of current density with sulphate ions. The comparative higher degradation in chloride media may be due to the formation of active chlorine species [12]. The decrease in degradation in the presence of anions is due to the scavenging effect of hydroxyl radicals by sulphate (Equation (9)) and chloride (Equations (10) and (11) forming hydroxyl ions [54]. In the presence of carbonates, the degradation of RhB is suppressed by the formation of FeCO$_3$. Jinisha et al. [36] concluded that the effect of anions like sulphates and carbonates does not influence the production of H$_2$O$_2$.

$$HO^{\bullet} + SO_4^{2-} \rightarrow OH^- + SO_4^- \tag{9}$$

$$HO^{\bullet} + Cl^- \leftrightarrow HOCl^- \tag{10}$$

$$HOCl^{-\bullet} + Cl^- \rightarrow Cl_2^- + OH^- \tag{11}$$

### 3.2.7. Radical Scavenging Tests

To evaluate the contribution of various radicals such as hydroxyl radical on the surface ($^{\bullet}$OH$_{surf}$) and on bulk ($^{\bullet}$OH$_{bulk}$), super oxides (O$_2$$^{-\bullet}$) and singlet oxygen ($^1$O$_2$) scavengers

such as t-butanol, KI, benzoquinone and sodium azide (NaN$_3$) were utilized. The scavenger t-butanol is known to have a good ability to scavenge $^\bullet$OH [12]. Benzoquinone was used as superoxide radical scavenger [61]. The surface $^\bullet$OH present on the catalyst surface are scavenged by KI [66]. The potential generation of singlet oxygen can be evaluated by sodium azide [67].

As shown in Figure 6a, the surface $^\bullet$OH activity is insignificant, whereas the bulk $^\bullet$OH radical strongly contributed to HEF activity. From Figure 6a, the non-radical pathway via singlet oxygen is the predominant pathway of HEF activity, and followed the trend, $^1O_2 > {^\bullet}OH_{bulk} > O_2^{-\bullet}$.

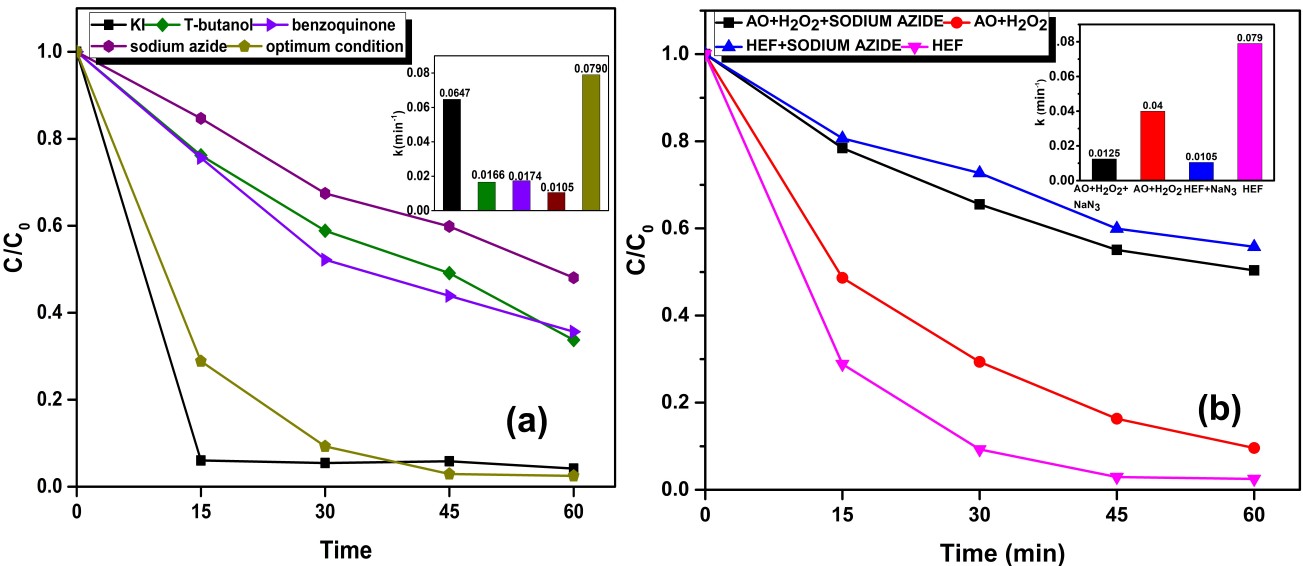

**Figure 6.** (**a**) Radical scavenging tests with rate of reactions; (**b**) sodium azide quenching experiments at different conditions.

To confirm the contribution of singlet oxygen in the system, quenching using NaN$_3$ was conducted in the presence and absence of the MnFe$_2$O$_4$–GO catalyst. As illustrated in Figure 6b, a strong presence of a non-radical pathway is evident in the AO + H$_2$O$_2$ system, where the NaN$_3$ lowered the RhB degradation rate to 0.0125 min$^{-1}$, comparable to the HEF with NaN$_3$ (0.0105 min$^{-1}$). This observation indicates the formation of $^1O_2$ is not solely from the catalyst.

Similarly, Lu et al. [68] reported the formation of singlet oxygen in the AO+H$_2$O$_2$ system in the presence of chloride (Cl$^-$). Herein, the chloride presence is evident, and the reasons might be (1) from the tap water used for making RhB-simulated water (37.7 mg L$^{-1}$ of chloride), (2) Cl$^-$ release from the RhB structure due to degradation. So, the singlet oxygen formation can be attributed to the singlet oxygen formation in chloride medium.

### 3.2.8. Role of Fe, Mn, GO and MnFe$_2$O$_4$

The individual contribution of Fe, Mn, GO and MnFe$_2$O$_4$ spinel structure in providing HEF activity were investigated using catalysts prepared under respective conditions by excluding other precursors involved. Because of the availability of numerous oxygen functional groups, large surface area, electrical conductivity and mechanical stability, graphene and GO are very promising support materials [69,70], and thereby enhance catalytic activity [71]. The enhancement in catalytic activity of the L–GO–ZnO composite on the photocatalytic activity of RhB was reported by Yaqoob et al. [10]. There are a few studies in the literature which report the catalytic activity of GO [72]. Thus, the role of GO alone is also examined to investigate the possibility of GO as a catalyst for HEF activity. Figure 7a depicts the different degradation rates of RhB obtained under each condition. From Figure 7a, it is clear that the catalyst works efficiently when it is in the form of spinel structure with GO support with a rate of RhB removal of 0.079 min$^{-1}$. In the case of GO, Mn,

Fe and MnFe$_2$O$_4$ without GO, the rates of RhB removal were 0.0449 min$^{-1}$, 0.0601 min$^{-1}$, 0.0610 min$^{-1}$, 0.0640 min$^{-1}$, respectively (inset of Figure 7a). The GO solely does not directly contribute to the HEF activity, as the RhB degradation in that case is similar to that of AO + H$_2$O$_2$ (0.04 min$^{-1}$). Similar results were obtained by Yao et al. [71]. In addition, this study observed an additional benefit of GO in the catalyst composite development, as the GO support contributed to lowering the solvothermal synthesis duration. In the absence of GO, MnFe$_2$O$_4$ development required a minimum of 9 h, whereas in the presence of GO, proper catalyst development was observed within 6 h of solvothermal treatment.

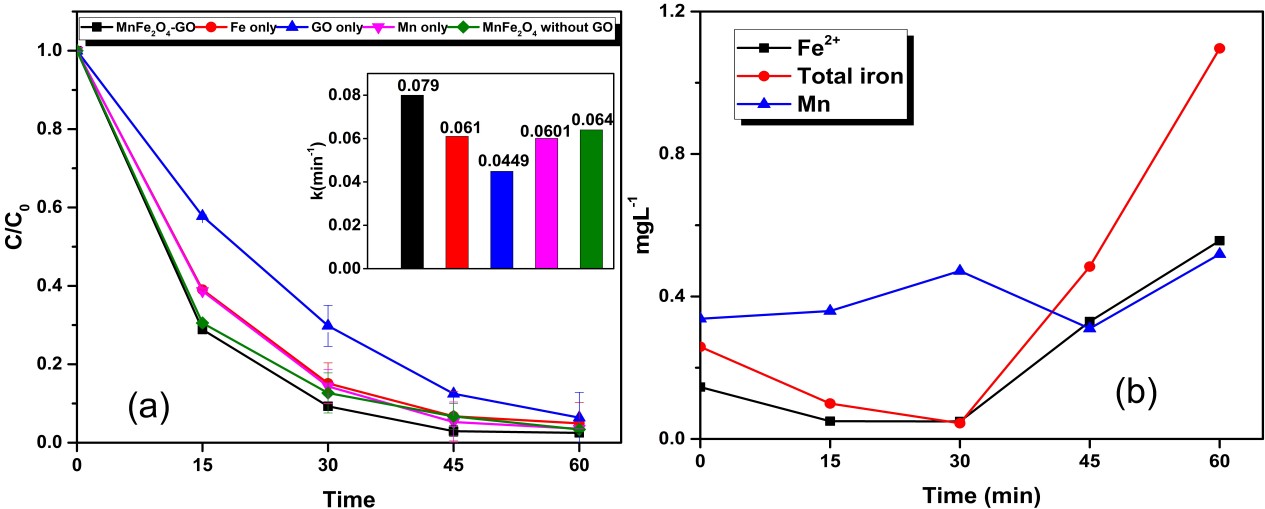

**Figure 7.** (**a**) Role of Fe, Mn and GO in catalyst with rate of reaction. (**b**) Leaching of Fe and Mn.

### 3.2.9. Leaching Studies

The stability of the as-synthesized catalyst was evaluated by analyzing the Fe and Mn leaching using ICP-OES and 1,10- phenanthroline method [39]. The amount of leached iron and Mn from the catalyst is illustrated in the Figure 7b. The amount of Fe$^{2+}$ and total iron in the initial 30 min showed no significant leaching. After 30 min, there was a gradual increase in concentration, indicating the leaching of Fe$^{2+}$ and total iron into the solution. The amount of Mn leached out showed fluctuating results which indicated that no gradual Mn release occurred.

### 3.2.10. Mechanism of Dye Removal

The mechanism involved in the HEF oxidation of RhB is given in Figure 8. The Fe$^{2+}$ from the catalyst decomposes in-situ generated H$_2$O$_2$ to form hydroxyl radicals (Equation (5)) which degrade the RhB molecules into CO$_2$, H$_2$O and other byproducts. Mn metal present in MnFe$_2$O$_4$ strongly accelerated the H$_2$O$_2$ decomposition, as in Equations (12) and (13) [34,35]. The Fe$^{2+}$ formation from Fe$^{3+}$ occurred by electron loss (which is explained by Equations (1)–(3)). All the possible reductions of Fe$^{3+}$ to Fe$^{2+}$ and degradation occur at the cathode, as RhB is a cationic dye [54].

$$Mn^{2+} + H_2O_2 \rightarrow Mn^{3+} + HO^{\bullet} + OH^{-} \tag{12}$$

$$Mn^{3+} + H_2O_2 \rightarrow Mn^{2+} + HO_2^{\bullet} + H^{+} \tag{13}$$

(RhB) degradation by reactive radicals is possible by ring-opening reaction and N-de-ethylation pathways [63]. As explained in Section 3.2.1, the slow decolorization of the RhB in AO + H$_2$O$_2$ system in the absence of catalyst may be facilitated by singlet oxygen. Singlet oxygen can be possibly formed by deprotonization, followed by electron rearrangement of oxygen molecules and disproportionation of H$_2$O$_2$ [73–75].

The results from this study compared with the efficiency of various catalysts in the removal of various dyes using HEF process, and given in Table 1. From Table 1, it is clear

that the use of $MnFe_2O_4$–GO composite catalyst in the present study has significantly reduced the total electrolysis time, while keeping higher dye removal efficiency. The efficiency of the process is higher when compared with the conventional EF process [13].

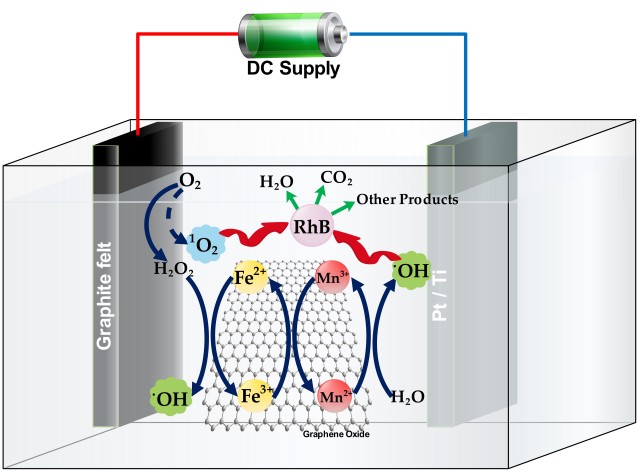

**Figure 8.** Mechanism of dye removal.

**Table 1.** Efficiency of various catalysts used for removal of dyes by electro-Fenton process.

| Catalyst Used | Dye Removed | Efficiency of the Catalyst | Electrodes Used | Electrolysis Time | Reference |
|---|---|---|---|---|---|
| $MnFe_2O_4$-GO | Rhodamine B | 97.51% colour removal | Pt/Ti and graphite felt | 60 min | Present study |
| $Fe^{2+}$ | Reactive yellow 160 | 91% colour removal | BDD and air diffusion cathode | 360 min | [50] |
| $Fe^{2+}$ | Alizarin red | >95% TOC removal | Graphite felt | 210 min | [13] |
| Magnetite | Rhodamine B | 97.3% colour removal | Graphite | 180 min | [54] |
| $Fe_3O_4$ nanoparticles | Methyl orange | 86.6% colour removal | $RuO_2$/Ti and C-PTFE | 90 min | [76] |
| Modified sponge iron | Rhodamine B | 99.71% colour removal | Pt sheet and GDE | 120 min | [37] |
| Iron doped SBA-15 mesoporous silica | Rhodamine B | 97.7% colour removal | Graphite | 180 min | [36] |
| $Fe_3O_4$/rGO | Reactive red 195 | 93.34% colour removal | Stainless steel rods | 60 min | [25] |
| Vermiculite | Ponceau SS | 92.4% colour removal | BDD and carbon PTFE air diffusion electrode | 360 min | [57] |

### 3.3. HEF Activity in Real Textile Wastewater

The wastewater collected was slightly acidic in nature (pH of 5.34) and had a conductivity of 13.2 mS $cm^{-1}$. The optimized experimental conditions (pH-3, applied voltage of 3 V, catalyst concentration of 20 mg $L^{-1}$) were applied to the textile wastewater for its effective treatment. The colour change was examined at a wavelength of 555 nm. As illustrated in Figure 9, colour reduction of 61.4% ± 0.061 was obtained after HEF treatment. The decolorization is caused by the breakdown of the dyes belonging to the chromogen group into its by-products [77]. Similarly, Nidheesh et al. [63] found 97.5% colour removal in textile wastewater by the HEF process. Colour removal in real textile water was investigated in batch and continuous modes by Nidheesh and Gandhimathi [77]. In the batch study, there was 83% colour removal and in the continuous study, 68% colour removal was obtained.

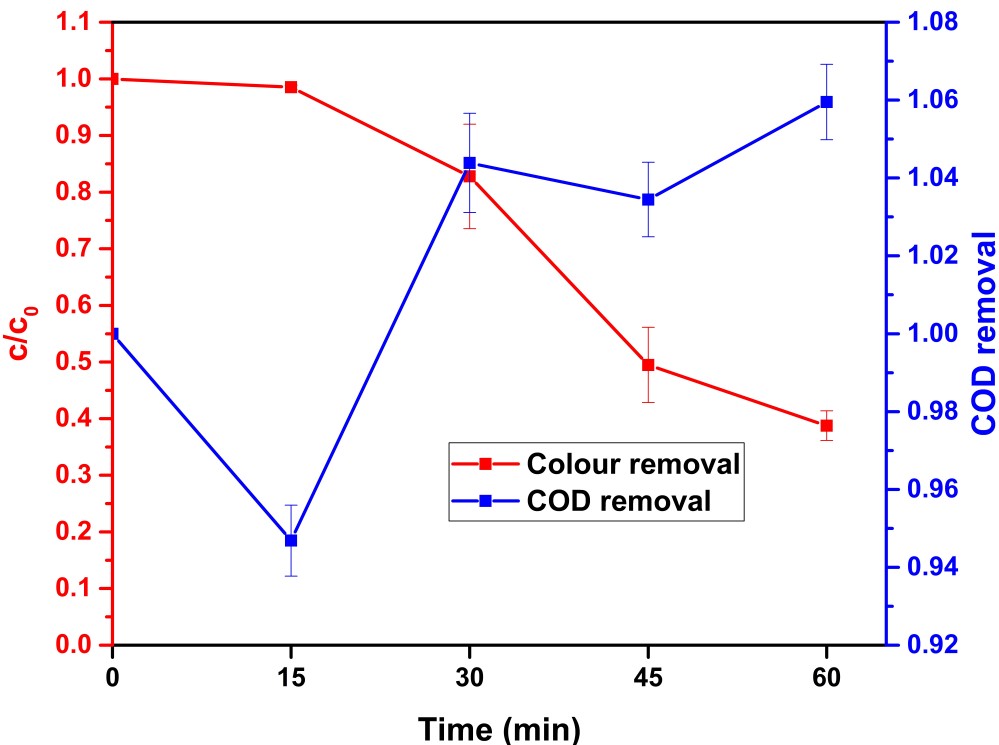

**Figure 9.** Colour removal and COD (mg L$^{-1}$) removal of real textile wastewater.

The COD were determined at different intervals to understand the degree of pollutant mineralization. The COD removal within 1 h is shown in Figure 9. Fluctuating COD percentage removal was observed. The abnormal result may be due to the mineralization of the compounds and intermediates present in the wastewater [78]. The degradation of complex organic pollutants, (non-detectable in COD analysis) to simpler organic pollutants detectable in COD analysis may result in rising COD values during the HEF process [77]. In general, the dyes present in textile wastewater are aromatic compounds with multiple aromatic rings and are not degradable by dichromate ions (used for COD analysis). However, these compounds are degradable by hydroxyl radicals, as they have higher oxidation potential than dichromate ions. However, the intermediate products and byproducts formed during the HEF process may not be as stable as that of dye and can be degraded by dichromate ions (thus, COD values will be provided by these compounds).

The TOC of the textile wastewater at different time intervals was determined, and the values before and after treatment are shown in Table 2. The initial TOC of 1551 mg L$^{-1}$ reduced to 1478 mg L$^{-1}$ after an electrolysis time of 60 min, with an increase in BOD from 631.58 to 1894.74 mg L$^{-1}$ (Table 2). The conversion of complex organic structures to simpler biodegradable organic compounds, instead of total mineralization, could end up in lower TOC removal with improvement in BOD [79,80]. Generally, carboxylic acids such as oxalic acids are formed as intermediates in the degradation of dyes [81]. These compounds are unable to be degraded with •OH due to their inefficiency [82], while microorganisms are found to be effective for the degradation of organic compounds [83]. This might have resulted in the increase in BOD.

Textile effluent characteristics before and after treatment were examined (Table 2). A slight increase in chloride and sulphate values was observed after the treatment. The increase in chloride and sulphate values may be due to the mineralization of chloride and sulphate containing complex aromatics present in the wastewater. Textile wastewaters are characterized by a high content of chloride in the range of 1600–2100 mg L$^{-1}$, as reported by the earlier studies [3]. Brillas et al. [5] suggested the increasing chances of chlorinated compounds in the wastewater after electrochemical processes. Studies have reported the increase in concentration of inorganics after electrochemical treatment [84].

For the treatment of textile effluents, hybrid systems are considered, as they will resolve the various limitations associated with the treatment using single systems. The increase in chloride and sulphate after HEF treatment may affect subsequent treatment. For example, AOPs followed by biological treatments are preferred because of the enhancement in biodegradability after the treatment [85]. However, the increase in chloride and sulphate can hamper the biological treatments [84]. Sodium and magnesium have been significantly decreased after treatment. The BOD/COD ratio shot up from 0.07 to 0.21, indicating the increase in biodegradability of the wastewater after treatment. Thus, the HEF process can be effectively used as a pre-treatment process for wastewater.

**Table 2.** Physicochemical characteristics of textile wastewater.

| Parameters | Before Treatment | After Treatment |
|---|---|---|
| BOD (mg $L^{-1}$) | 631.58 | 1894.74 |
| COD (mg $L^{-1}$) | 8521.81 | 9028.43 |
| TOC (mg $L^{-1}$) | 1551 | 1478 |
| Chloride (mg $L^{-1}$) | 3830.12 | 4080.74 |
| Sulphate (mg $L^{-1}$) | 3936.71 | 5111.11 |
| Sodium (mg $L^{-1}$) | 3939 | 3631 |
| Magnesium (mg $L^{-1}$) | 121 | 14 |

## 4. Conclusions and Future Perspectives

Solvothermally synthesized $MnFe_2O_4$ supported on GO was used as the catalyst for the effective removal of RhB from aqueous solution by HEF process. The XRD, FTIR and TEM confirmed the formation of $MnFe_2O_4$ over GO. The optimized experimental conditions were found as pH 3, applied voltage of 3 V, catalyst concentration of 20 mg $L^{-1}$ and initial pollutant concentration of 10 ppm. After an electrolysis time of 60 min, a removal efficiency of 97.51% could be achieved with these optimized conditions. Bulk $^\bullet OH$ and super-oxides were found out to be the superior oxidants controlling the process. The role of GO in catalyst was significant, as the time for the catalyst formation could be reduced. Mn leaching from the catalyst was found to be insignificant. However, the leaching of iron could be seen after an electrolysis time of 30 min. The optimized experimental conditions were applied to the real textile wastewater and an efficient colour removal of 61.23% at 555 nm could be observed. A TOC removal of 4.71% was obtained. Even though the COD results were fluctuating, an increase in BOD/COD ratio from 0.07 to 0.21 was observed after treatment, which indicated that the biodegradability of the wastewater was significantly enhanced. Thus, the HEF process could be used as an effective pre-treatment method for wastewaters.

The main challenge faced during the HEF process was the need for frequent replacement of the cathode, especially for the treatment of real wastewater. The efficiency of the dye removal greatly depends on the cathode, and passivation of the cathode during treatment will negatively affect its efficiency. Another challenge faced is associated with the variation in the behavioural properties of the catalyst during multiple synthesis.

The need for pilot-scale research in textile wastewater has to be mentioned. Further studies require an increase in the efficiency of textile wastewater in terms of colour and COD, as well as TOC. The degradation of textile wastewater can also be found as a factor of time, i.e., whether the optimized conditions will increase efficiency in extending the electrolysis time. Biological studies can be conducted to confirm the biodegradability of the textile wastewater after treatment.

**Author Contributions:** Conceptualization, P.V.N.; methodology, P.V.N. and J.S.; investigation, G.A. and J.S.; writing—original draft preparation, G.A.; writing—review and editing, P.V.N. and G.A.; supervision, P.V.N. All authors have read and agreed to the published version of the manuscript.

**Funding:** This research received no external funding.

**Institutional Review Board Statement:** Not applicable.

**Informed Consent Statement:** Not applicable.

**Data Availability Statement:** Data is contained within the article.

**Acknowledgments:** The authors are thankful to the Director, CSIR NEERI, for their support. The authors would also like to thank the Sophisticated Analytical Instrument Facility (SAIF), Indian Institute of Technology-Bombay, India; Indian Science Technology and Engineering Facilities Map (I-STEM), the Centre for Nano Science and Engineering, Indian Institute of Science, Bangalore; and the Institute Instrumentation Centre (IIC), the Indian Institute of Technology, Roorkee, India, for their assistance with various analyses.

**Conflicts of Interest:** The authors declare no conflict of interest.

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
