# Peer review of "Removal of Synthetic Dye from Aqueous Solution Using MnFe2O4-GO Catalyzed Heterogeneous Electro-Fenton Process"

_water, doi:10.3390/w14203350_

Round 1

Reviewer 1 Report

This work used MnFe2O4 -GO catalyzed Heterogeneous electro -Fenton process to remove Rhodamine B (RhB) dye from aqueous solution. The effect of various operational parameters like pH, applied voltage, catalyst concentration, initial pollutant concentration and effect of ions were investigated. This work may give some suggestions on dyeing waster water treatment. However, the significantce of this study is not clear. Only removing Rhodamine B (RhB) is not enough for the dyeing water water treatment. Moreover, the BOD, COD and TOC values were increased after treatment, this is unacceptable. So this study should be redesigned and investigated further.

Reviewer 2 Report

Dear editor,

Thanks for inviting me to review this manuscript titled “Removal of Synthetic dye from aqueous solution using MnFe2O4-GO catalyzed Heterogeneous electro-Fenton process”. HEF process using MnFe2O4-GO catalyst was used to remove RhB from aqueous solutions. The effect of various operational parameters were investigated and the results presented practical and logical application potential. Thus I recommend this manuscript be considered for publication after some revisions.

--In section 1, the author mentioned various transition metals such as cobalt, copper, nickel, vanadium and manganese have been utilized as solid catalysts with iron/iron oxides, but it is lack a statement of why manganese was chosen in this study. The advantages of manganese as a solid catalyst should be added.

-- In 3.1, only the characterization of MnFe2O4-GO was analyzed, and the comparison of GO only (both FeCl3 and MnCl2 was not added) and MnFe2O4 without support (GO was not added) mentioned above should be added to highlight the catalytic advantages of the composite.

-- The format of headings should be uniformed.

--In 3.2.9, what is the reason for the leaching of Fe2+ and why is Mn2+ not leached? Is there any consideration of how to prevent leaching?

--In rows of 435-436, what is the increase in chlorine and sulfate values of the treated textile wastewater? Does it make the subsequent treatment more difficult?

--In 3.2, the effect of degradation time and pollutant concentration on degradation performance should be added.

-- There were some formatting errors in the manuscript (e.g., Ref. 22) that authors should checked.

--A depth mechanism should be discussed.

Author Response

Thanks for inviting me to review this manuscript titled “Removal of Synthetic dye from aqueous solution using MnFe2O4-GO catalyzed Heterogeneous electro-Fenton process”. HEF process using MnFe2O4-GO catalyst was used to remove RhB from aqueous solutions. The effect of various operational parameters was investigated and the results presented practical and logical application potential. Thus, I recommend this manuscript be considered for publication after some revisions.

Response: Thank you very much for supporting our work.

  1. In section 1, the author mentioned various transition metals such as cobalt, copper, nickel, vanadium and manganese have been utilized as solid catalysts with iron/iron oxides, but it is lack a statement of why manganese was chosen in this study. The advantages of manganese as a solid catalyst should be added.

Response: Changes have been incorporated in the introduction section of the revised manuscript. Please refer page 2 lines 83-91.

  1. In 3.1, only the characterization of MnFe2O4-GO was analyzed, and the comparison of GO only (both FeCl3 and MnCl2 was not added) and MnFe2O4 without support (GO was not added) mentioned above should be added to highlight the catalytic advantages of the composite.

Response: Thank you for this suggestion. The conversion of graphite powder to graphene oxide (GO) followed modified Hummer’s method. The detailed characterization studies for confirming GO formation, solvothermal conversion of metal chlorides to metal oxides were already reported in the previous studies of our group (Scaria and Nidheesh, 2022).

  1. The format of headings should be uniformed.

     Response: The headings have been formatted in the modified version of the manuscript.

  1. In 3.2.9, what is the reason for the leaching of Fe2+ and why is Mn2+ not leached? Is there any consideration of how to prevent leaching?

Response: Thank you for this suggestion. The leaching study was revaluated and the modified data is provided in figure 7 (b). Please refer page 12 line 447- 453.

  1. In rows of 435-436, what is the increase in chlorine and sulfate values of the treated textile wastewater? Does it make the subsequent treatment more difficult?

     Response: Thank you for the comment. The explanation to this suggestion has been included in the revised version of the manuscript. Page 15 line 522-531

  1. In 3.2, the effect of degradation time and pollutant concentration on degradation performance should be added.

      Response: Thank you for the comment. The effect of degradation time and effect of pollution concentration have been discussed. Please refer section 3.2.5.

  1. There were some formatting errors in the manuscript (e.g., Ref. 22) that authors should checked.

     Response: Thank you for your constructive comment. The errors have been checked and corrected on the modified version of the manuscript.

  1. A depth mechanism should be discussed.

Response: Thank you for your constructive comment. The degradation mechanism of RhB has been added in the revised version of the manuscript. Please refer section 3.2.10.

Reviewer 3 Report

Greetings, Editor thank you for providing me with the opportunity to review the article. I reviewed the article with water-1950921. Overall, the article structure and content are suitable for the WATER journal. I am pleased to send you major level comments, there are some serious flaws which need to be corrected before publication. Please consider these suggestions as listed below.

  1. The title seems ok.
  2. The abstract seems to be good. Please add one more introductory line of your objective in beginning of abstract.
  3. Research gap should be delivered on more clear way with directed necessity for the future research work.
  4. Remove `` MnFe2O4-GO`` from keywords.
  5. Introduction section must be written on more quality way, i.e., more up-to-date references addressed. Please target the specific gap such as 2015-2021 etc.
  6. The novelty of the work must be clearly addressed and discussed, compare previous research with existing research findings and highlight novelty.
  7. What is the main challenge?
  8. In introduction Page 1 Line 27 need another reference please cite this article here- Role of nanomaterials in the treatment of wastewater: A review.
  9. The main objective of the work must be written on the more clear and more concise way at the end of introduction section.
  10. Page 1 Line 35 please cite this article``Graphene oxide–ZnO nanocomposite: An efficient visible light photocatalyst for degradation of rhodamine B``
  11. Similar Page 2 Line 82 need another reference please cite- Hybrid nanocomposites based on graphene and its derivatives: from preparation to applications.
  12. Please check the abbreviations of words throughout the article. All should be consistent.
  13. Please include all chemical/instrumentation brand name and other important specification.
  14. Please provide space between number and units. Please revise your paper accordingly since some issue occurs on several spots in the paper.
  15. Overall result section is well explained.
  16. Regarding the replications, authors confirmed that replications of experiment were carried out. However, these results are not shown in the manuscript, how many replicated were carried out by experiment? Results seem to be related to a unique experiment. Please, clarify whether the results of this document are from a single experiment or from an average resulting from replications. If replicated were carried out, the use of average data is required as well as the standard deviation in the results and figures shown throughout the manuscript. In case of showing only one replicate explain why only one is shown and include the standard deviations.
  17. Please add a comparative discussion section. It would be more better for reader.
  18. Section 4 should be renamed by Conclusion and Future perspectives. Conclusion section is missing some perspective related to the future research work, quantify main research findings, highlight relevance of the work with respect to the field aspect.
  19. To avoid grammar and linguistic mistakes, moderate level English language should be thoroughly checked. Please revise your paper accordingly since several language issue occurs on several spots in the paper.
  20. Reference formatting need carefully revision. All must be consistent in one formate. Please follow the journal guidelines.

Author Response

Greetings, Editor thank you for providing me with the opportunity to review the article. I reviewed the article with water-1950921. Overall, the article structure and content are suitable for the WATER journal. I am pleased to send you major level comments, there are some serious flaws which need to be corrected before publication. Please consider these suggestions as listed below.

Response: Thank you very much for supporting our work.

  1. The title seems ok.

Response: Thank you so much sir.

  1. The abstract seems to be good. Please add one more introductory line of your objective in beginning of abstract.

Response: Thank you. as per your suggestion objective has been added in the abstract. Please refer page 1 line11-12.

  1. Research gap should be delivered on more clear way with directed necessity for the future research work.

Response: The research gap identified and the importance of this study is explained in the revised manuscript (page 2 line 92- 104). The future research work with mention to challenges faced have been added in the revised version of the manuscript. Please refer section 4.

  1. Remove `` MnFe2O4-GO`` from keywords.

Response: As per your suggestion it has been removed from the Keywords.

  1. Introduction section must be written on more quality way, i.e., more up-to-date references addressed. Please target the specific gap such as 2015-2021 etc.

Response: Thank you for this suggestion. The introduction part is revised as per the suggestion to discuss on the articles in the specific time gap. Only the relevant scientific explanations have been taken from articles earlier to this time gap.

  1. The novelty of the work must be clearly addressed and discussed, compare previous research with existing research findings and highlight novelty.

Response: As per your suggestion changes has been made in the modified version of the manuscript. Please refer page 2 line 92 -104

  1. What is the main challenge?

Response: The challenges faced are included in the revised version of the manuscript. Please refer section 4.

  1. In introduction Page 1 Line 27 need another reference please cite this article here- Role of nanomaterials in the treatment of wastewater: A review.

Response: As per your suggestion this article has been cited in the revised version of the manuscript.

  1. The main objective of the work must be written on the more clear and more concise way at the end of introduction section.

Response: As per your suggestion, the objective has been added in the introduction section. Please refer page 1 line 99 to 104.

  1. Page 1 Line 35 please cite this article``Graphene oxide–ZnO nanocomposite: An efficient visible light photocatalyst for degradation of rhodamine B``

Response: As per your suggestion this article has been cited in the revised version of the manuscript.

  1. Similar Page 2 Line 82 need another reference please cite- Hybrid nanocomposites based on graphene and its derivatives: from preparation to applications.

Response: As per your suggestion, the mentioned article has been cited in the revised version of the manuscript.

  1. Please check the abbreviations of words throughout the article. All should be consistent.

Response: The abbreviations have been thoroughly checked and changes have been made wherever needed in the modified version of the manuscript.

  1. Please include all chemical/instrumentation brand name and other important specification.

Response: The brand names and specification of chemicals and instruments have been added in the revised version of the manuscript.

  1. Please provide space between number and units. Please revise your paper accordingly since some issue occurs on several spots in the paper.

Response: Changes has been incorporated in the revised version of the manuscript.

  1. Overall result section is well explained.

Response: Thank you so much for your valuable comment.

  1. Regarding the replications, authors confirmed that replications of experiment were carried out. However, these results are not shown in the manuscript, how many replicated were carried out by experiment? Results seem to be related to a unique experiment. Please, clarify whether the results of this document are from a single experiment or from an average resulting from replications. If replicated were carried out, the use of average data is required as well as the standard deviation in the results and figures shown throughout the manuscript. In case of showing only one replicate explain why only one is shown and include the standard deviations.

Response: All the Fenton experiments were carried out in replications and the average of the replicates was taken to express the data. The graphs are expressed with standard deviations. The figures have also been added with standard deviations except for the experiments of scavenging and characterization of wastewater before and after treatment.

  1. Please add a comparative discussion section. It would be better for reader.

 Response: A comparative table showing the efficiencies of various catalyst with catalyst used in the study has been inserted in the revised version of the manuscript. Please refer Table 1 and explanation in page 13 line 476-480.

  1. Section 4 should be renamed by Conclusion and Future perspectives. Conclusion section is missing some perspective related to the future research work, quantify main research findings, highlight relevance of the work with respect to the field aspect.

Response: As per the suggestion, the section has been renamed and future research works have been highlighted.

  1. To avoid grammar and linguistic mistakes, moderate level English language should be thoroughly checked. Please revise your paper accordingly since several language issue occurs on several spots in the paper.

Response: As per the suggestion, manuscript revised with thorough checking to avoid grammatical and linguistic errors.

  1. Reference formatting need carefully revision. All must be consistent in one format. Please follow the journal guidelines.

Response: Thank you for the comment. The references are re-checked to avoid formatting errors.

Reviewer 4 Report

The submitted manuscript (No. water-1950921) investigated the removal of synthetic dye from aqueous solution using MnFe2O4-GO catalyzed heterogeneous electro-Fenton process. The effect of various operational parameters like pH, applied voltage, catalyst concentration, initial pollutant concentration and effect of ions were investigated in detail. Quenching tests demonstrated that bulk hydroxyl radical and super-oxides were the primary oxidants for RhB degradation. The proposed MnFe2O4-GO catalyzed heterogeneous electro-Fenton process was successfully applied to treat the real textile wastewater with the efficient color removal and the great increase in BOD/COD ratio. Overall, the manuscript was meaningful and practicable. Nevertheless, there are still several problems to be solved before the manuscript is accepted by Water.

Specific comments:

1) The first letter of the words of “Synthetic” and “Heterogeneous” in Title should be written in lower case. There were redundant spaces between number and name for “2.7 Energy consumption”, “2.8 Leaching of catalyst”, “3.2.1 Comparison between different processes”, “3.2.2 Effect of catalyst dosage” and so on.

2)  Some equations were not aligned well, for example Equation 11. Similarly, some figures were also not aligned well, such as Figure 4, Figure 5 and Figure 7. The legend of Figure 5 was also difficult to distinguish clearly.

3) Figure 2 (c) demonstrated the effect of solution pH in the range of 2-11 on the degradation of RhB. The removal efficiency of RhB at pH 3 reached 97.72%, while only 18.08% of RhB was removed at pH 7. It is necessary to carry out the experiment of RhB degradation in the pH range of 3-7, for example pH 5.

4) The concentration of H2O2 was a crucial factor for the generation of reactive oxidants and the degradation of RhB. However, the information about the determination method of H2O2 and the change of H2O2 concentration was not found in the manuscript. The concentration of H2O2 could be accurately determined by the previously-reported ABTS method (Multi-wavelength spectrophotometric determination of hydrogen peroxide in water with peroxidase-catalyzed oxidation of ABTS. Chemosphere, 2018. DOI: 10.1016/j.chemosphere.2017.11.091).

5) Figure 7 showed that the percentage of COD removal was fluctuating in the narrow range of 0.94-1.06. So, the description of the sentence “The percentage of COD removal with increase in time is shown in Figure 7.” in Line 426, Page 11 was incorrect.

6)  Quenching tests demonstrated that singlet oxygen was the predominant oxidant for RhB degradation. The conclusion was very important for this manuscript. However, only NaN3 was used as the quenching agent. Previous reports (such as Chemical Engineering Journal, 2022, DOI: 10.1016/j.cej.2021.132438 and Water Research, 2022, DOI: 10.1016/j.watres.2022.119095) had also used furfuryl alcohol as the quenching agent to identify the contribution of singlet oxygen on organic contaminants. Additionally, Previous reports (such as Chemical Engineering Journal, 2022, DOI: 10.1016/j.cej.2021.132438 and Water Research, 2022, DOI: 10.1016/j.watres.2022.119095) also found that the reaction between molecular form and deprotonated form of oxidant was an important pathway to generate singlet oxygen. So, singlet oxygen might be generated from the reaction between H2O2 and its deprotonated form (i.e., HO2).

Author Response

The submitted manuscript (No. water-1950921) investigated the removal of synthetic dye from aqueous solution using MnFe2O4-GO catalyzed heterogeneous electro-Fenton process. The effect of various operational parameters like pH, applied voltage, catalyst concentration, initial pollutant concentration and effect of ions were investigated in detail. Quenching tests demonstrated that bulk hydroxyl radical and super-oxides were the primary oxidants for RhB degradation. The proposed MnFe2O4-GO catalyzed heterogeneous electro-Fenton process was successfully applied to treat the real textile wastewater with the efficient color removal and the great increase in BOD/COD ratio. Overall, the manuscript was meaningful and practicable. Nevertheless, there are still several problems to be solved before the manuscript is accepted by Water.

Response: Thank you very much for supporting our work.

  1. The first letter of the words of “Synthetic” and “Heterogeneous” in Title should be written in lower case. There were redundant spaces between number and name for “2.7 Energy consumption”, “2.8 Leaching of catalyst”, “3.2.1 Comparison between different processes”, “3.2.2 Effect of catalyst dosage” and so on.

Response:  As per your suggestion the title has been modified and the spaces between the mentioned headings have been removed.

  1. Some equations were not aligned well, for example Equation 11. Similarly, some figures were also not aligned well, such as Figure 4, Figure 5 and Figure 7. The legend of Figure 5 was also difficult to distinguish clearly.

Response: Equations have been formatted. Figures have also been aligned. Figure 6 ( as per revised version of the manuscript) has been reinserted in the revised version of the manuscript.

  1. Figure 2 (c) demonstrated the effect of solution pH in the range of 2-11 on the degradation of RhB. The removal efficiency of RhB at pH 3 reached 97.72%, while only 18.08% of RhB was removed at pH 7. It is necessary to carry out the experiment of RhB degradation in the pH range of 3-7, for example pH 5.

Response: As shown in the data higher efficiency was obtained when the pH of the system was in the range of 2-3. Higher degradation was shown by pH 3. The degradation decreased as increased the pH to neutral condition and then further decreased on applying alkaline conditions. This trend is in alignment with the previous reports. So, this study chosen an acidic pH (pH 2, pH 3), neutral pH (7), and alkaline pH (pH 9, pH 11) conditions as model conditions to evaluate the effect of pH.

Many studies reported the higher degradation of compounds at pH of 3. Zheng et al., (2021) observed a high degradation in RhB at pH of 3 on using NiFe2O4/Fe2O3 as the heterogenous catalyst with graphite felt as the electrodes. Nidheesh et al., (2014) studied the degradation of RhB using magnetite as the catalyst and graphite as the electrode. The authors also obtained higher efficiency at the pH of 3. Fayazi and Ghanei-Motlagh, (2020) obtained higher degradation efficiency of methylene blue at pH 3 when sepiolite/pyrite composite was used as the catalyst and graphite and platinum sheet was used as the electrodes. On the degradation of Ponceau SS dye using heterogenous electro Fenton process, dos Santos et al., (2020) observed that at pH 3 higher degradation was obtained when vermiculite was used as the catalyst and BDD and air – diffusion cathode was used.

  1. The concentration of H2O2was a crucial factor for the generation of reactive oxidants and the degradation of RhB. However, the information about the determination method of H2O2 and the change of H2O2 concentration was not found in the manuscript. The concentration of H2O2 could be accurately determined by the previously-reported ABTS method (Multi-wavelength spectrophotometric determination of hydrogen peroxide in water with peroxidase-catalyzed oxidation of ABTS. Chemosphere, 2018. DOI: 10.1016/j.chemosphere.2017.11.091).

Response: Thank you for your valuable suggestion. In this study, no external addition of H2O2 was done in the system as the H2O2 was generated in-situ in the system. The quantitative determination of in-situ generated H2O2 follows spectrophotometric methods at wavelength of 240 nm which are limited in the current scenario as colored dyes are the chosen pollutant, which could interfere with the H2O2 quantification.

  1. Figure 7 showed that the percentage of COD removal was fluctuating in the narrow range of 0.94-1.06. So, the description of the sentence “The percentage of COD removal with increase in time is shown in Figure 7.” in Line 426, Page 11 was incorrect.

Response: The sentence has been corrected in the revised version of the manuscript. Please refer page 14 line 497.

  1. Quenching tests demonstrated that singlet oxygen was the predominant oxidant for RhB degradation. The conclusion was very important for this manuscript. However, only NaN3 was used as the quenching agent. Previous reports (such as Chemical Engineering Journal, 2022, DOI: 10.1016/j.cej.2021.132438 and Water Research, 2022, DOI: 10.1016/j.watres.2022.119095) had also used furfuryl alcohol as the quenching agent to identify the contribution of singlet oxygen on organic contaminants. Additionally, Previous reports (such as Chemical Engineering Journal, 2022, DOI: 10.1016/j.cej.2021.132438 and Water Research, 2022, DOI: 10.1016/j.watres.2022.119095) also found that the reaction between molecular form and deprotonated form of oxidant was an important pathway to generate singlet oxygen. So, singlet oxygen might be generated from the reaction between H2O2 and its deprotonated form (i.e., HO2

Response: The contribution of singlet oxygen can be evaluated by the quenching studies using sodium azide (NaN3) and furfuryl alcohol. Many literatures reported the effectiveness of both scavengers. NaN3 is chosen for this study, as per our previous studies reported. The mentioned articles have been cited in the revised version of the manuscript.

Reviewer 5 Report

General comments:

In this study, the authors successfully synthesized MnFe2O4-GO catalyst, and applied it and electron-Fenton process to degrade rhodamine B. The results found that the removal efficiency of 97.51% could be achieved with the optimized conditions after an electrolysis time of 60 min. In addition, bulk ●OH and super-oxides were found out to be the superior oxidants in these reaction process. Besides, this method plays an important role in increasing the biodegradability of real textile wastewater, and it can be utilized as the pre-treatment method for real wastewater. After careful review, this manuscript is well organized with enough data presentations. However, some minor errors (e.g., grammar errors) should be modified to improve the quality of this article. Therefore, minor revision of the manuscript is needed before it can be recommended for publication.

Specific comments:

Line 20: The increased COD value could not reflect the pollutant mineralization.

Lines 95, 96, 105, and 106: The dot in chemical formula should be in the middle position.

Line 116: Please check the predicate verb.

Line 144: The blank before and behind hyphen should be deleted.

Line 148: Please check the unit of temperature.

Line 161: The full stop is missing behind the bracket.

Line 165: Arabic numerals cannot appear at the beginning of a sentence.

Line 168: A blank is missing between “and” and “0.1 N”.

Line 188: The full name of TOC should be listed when TOC appeared at the first time.

Line 227: Please supplement the full stop.

Line 228: The spectra is plural, and please check the predicate verb.

Line 230: Please check the superscript.

Line 260: The word “if” should be changed into “of”.

Line 314: Please supplement the full stop.

Line 374: Please supplement a comma.

Line 419: Why choose 555nm as the test wavelength?

Line 443: Please supplement the unit of COD removal.

Line 453: Please check the predicate verb.

Line 458: Where is 4.71%? In Figure 7, the biggest COD removal efficiency should be around 1.06%.

Line 549: Please supplement the page number.

Author Response

In this study, the authors successfully synthesized MnFe2O4-GO catalyst, and applied it and electron-Fenton process to degrade rhodamine B. The results found that the removal efficiency of 97.51% could be achieved with the optimized conditions after an electrolysis time of 60 min. In addition, bulk ●OH and super-oxides were found out to be the superior oxidants in these reaction process. Besides, this method plays an important role in increasing the biodegradability of real textile wastewater, and it can be utilized as the pre-treatment method for real wastewater. After careful review, this manuscript is well organized with enough data presentations. However, some minor errors (e.g., grammar errors) should be modified to improve the quality of this article. Therefore, minor revision of the manuscript is needed before it can be recommended for publication.

Response: Thank you very much for supporting our work.

  1. Line 20: The increased COD value could not reflect the pollutant mineralization.

Response:  The abnormal trend in COD can be attributed to the breakdown of various other dye and chemical compounds present in the complex textile wastewater. More discussions on this was provided in the revised manuscript. The statement in line 20 page 1 has been modified in the revised manuscript.

  1. Lines 95, 96, 105, and 106: The dot in chemical formula should be in the middle position.

Response: Changes have been made in the revised manuscript. Please refer page 3 line 108-110.

  1. Line 116: Please check the predicate verb.

Response: Correction has been made. Please refer page 3 line 120.

  1. Line 144: The blank before and behind hyphen should be deleted.

Response: Correction has been made. Please refer page 4 line 158.

  1. Line 148: Please check the unit of temperature.

Response: Correction has been made. Please refer page 4 line 162.

  1. Line 161: The full stop is missing behind the bracket.

Response: It has been added. Please refer page 4 line 176.

  1. Line 165: Arabic numerals cannot appear at the beginning of a sentence.

Response: The sentence has been changed as per the suggestion. Please refer page 4 line 180.

  1. Line 168: A blank is missing between “and” and “0.1 N”.

Response: Correction has been made. Please refer page 4 line 183

  1. Line 188: The full name of TOC should be listed when TOC appeared at the first time.

Response: The full name has been enlisted in the revised version of the manuscript. Please refer page 5 line 205.

  1. Line 227: Please supplement the full stop.

Response: Change has been made in the revised version of the manuscript.

  1. Line 228: The spectra is plural, and please check the predicate verb.

Response: Changes have been made in the revised version of the manuscript. Please refer page 5 line 241 and page 6 line 248.

  1. Line 230: Please check the superscript.

Response: Change has been made. Please page 5 line 243

  1. Line 260: The word “if” should be changed into “of”.

Response: Change has been made in the revised version of the manuscript. Please refer page 7 line 273.

  1. Line 314: Please supplement the full stop.

Response: Change has been made in the revised version of the manuscript.

  1. Line 374: Please supplement a comma.

Response: Change has been made in the revised version of the manuscript.

  1. Line 419: Why choose 555nm as the test wavelength?

Response: 555 nm is the peak wavelength of RhB and is a characteristic of this dye. Similar bands at 555 nm for RhB was also obtained by Nidheesh and Rajan, (2016) and Fan et al., (2010).

  1. Line 443: Please supplement the unit of COD removal.

Response: The unit of COD removal has been added. Please refer page 15 line 537.

  1. Line 453: Please check the predicate verb.

Response: Change has been made. Please refer page 16 line 547.

  1. Line 458: Where is 4.71%? In Figure 7, the biggest COD removal efficiency should be around 1.06%.

Response: Figure 9 (as per the revised version of the manuscript) shows the data of COD and colour removal. The 4.71% is the percentage of TOC removal that we have obtained after the treatment.

  1. Line 549: Please supplement the page number.

Response: Page number has been added in the revised version of the manuscript. Please refer page 19 line 663.

Reviewer 6 Report

This paper is a somewhat innovative paper, suggesting:

(1) In the second paragraph of the introduction, it is suggested to add a few words to the current mainstream process of printing and dyeing wastewater treatment, which will help readers to have a better understanding of the context of the research.

(2) Please judge and explain whether the dye will be removed due to the change of pH during pH adjustment.

(3) How is the added catalyst separated from water? How to achieve reuse?

(4) It is recommended to add a schematic diagram of the device. Is there a photo of the electrodes? If there is, please add it so that readers can have a better understanding of the paper.

(5) In 3.2.6. Effect of ions

OH +??????−∙ (10)

???? •+????2+?? (11)

It is recommended to refer to the paper(Shiwei Gao, Zheng Wang, Haoran Wang, Yannan Jia, Nannan Xu, Xue Wang,Jiahao Wang, Chenyue Zhang, Tian Tian, Wei Shen. Peroxydisulfate activation using B-doped biochar for the degradation of oxytetracycline in water. Applied Surface Science,2022, https://doi.org/10.1016/j.apsusc.2022.153917) to optimize the writing of the formula, and make a better analysis of the relevant content in 3.2.7. Radical scavenging tests to improve the accuracy of the paper.

(6) In the real wastewater treatment experiment, the BOD increased a lot after the reaction, but the TOC changed little before and after the reaction. Why did the TOC change little, but the BOD increased greatly. Where does the big increase in BOD come from? ? Please analyze it.

Author Response

This paper is a somewhat innovative paper, suggesting:

Response: Thank you very much for supporting our work.

  1. In the second paragraph of the introduction, it is suggested to add a few words to the current mainstream process of printing and dyeing wastewater treatment, which will help readers to have a better understanding of the context of the research.

Response: As per your suggestion, a few more additions have been made in the introduction. The additions are highlighted in red.

  1. Please judge and explain whether the dye will be removed due to the change of pH during pH adjustment.

Response: No change in dye concentration with changes in solution pH was observed.

  1. How is the added catalyst separated from water? How to achieve reuse?

Response: The synthesized catalyst possessed magnetic properties. These magnetic properties will enhance the reusability as the catalyst can be separated using magnetic separation. The magnetic properties of MnFe2O4 composite was confirmed by Shao et al., (2012) and Goodarz Naseri et al., (2012). Further studies are required to confirm the reusability of the synthesized catalyst.

  1. It is recommended to add a schematic diagram of the device. Is there a photo of the electrodes? If there is, please add it so that readers can have a better understanding of the paper.

Response: We completely agree with your suggestion. The schematic diagram of the experimental setup has been added in the revised version of the manuscript. Please refer section 2.5.

  1. In 3.2.6. Effect of ions
  • OH +??−→????−∙ (10)

????− •+??−→??2−+??− (11)

It is recommended to refer to the paper (Shiwei Gao, Zheng Wang, Haoran Wang, Yannan Jia, Nannan Xu, Xue Wang,Jiahao Wang, Chenyue Zhang, Tian Tian, Wei Shen. Peroxydisulfate activation using B-doped biochar for the degradation of oxytetracycline in water. Applied Surface Science,2022, https://doi.org/10.1016/j.apsusc.2022.153917) to optimize the writing of the formula, and make a better analysis of the relevant content in 3.2.7. Radical scavenging tests to improve the accuracy of the paper.

Response: Thank you for the recommendation. As mentioned, corrections have been made and the suggested paper has been cited.

  1. In the real wastewater treatment experiment, the BOD increased a lot after the reaction, but the TOC changed little before and after the reaction. Why did the TOC change little, but the BOD increased greatly. Where does the big increase in BOD come from?? Please analyze it.

Response: Thank you for the comment. The explanation for this observation is provided in the revised manuscript (page 14 line 512 - 518).

Round 2

Reviewer 1 Report

The effect of GO on the treatment should be clearly discussed.

Author Response

Reviewer #1

1.The effect of GO on the treatment should be clearly discussed.

Response: Thank you so much for the constructive comment. The section 3.2.8 has been modified as per your suggestion in the revised version of the manuscript.

Reviewer 3 Report

Accepted 

Author Response

Accepted 

Response: Thank you very much for your support. 

Reviewer 4 Report

The authors have responsed my questions carefully.

Author Response

1. The authors have responsed my questions carefully.

Response: Thank you very much for your kind support.